

# Sucrose-delaying flower color fading associated with delaying anthocyanin accumulation decrease in cut chrysanthemum

Xiao-fen Liu[1,2,*], Ruping Teng[1,2,*], Lili Xiang[1,3], Fang Li[1,2] and Kunsong Chen[1,2]

[1] Zhejiang University, College of Agriculture and Biotechnology, Zijingang Campus, Hangzhou, China
[2] Zijingang Campus, Zhejiang University, Zhejiang Provincial Key Laboratory of Horticultural Plant Integrative Biology, Hangzhou, China
[3] Hunan Agricultural University, College of Horticulture, Changsha, China
* These authors contributed equally to this work.

## ABSTRACT

As fresh ornamental crops, vase life and post-harvested quality of cut flowers have attracted much attention. Flower color fading is the prominent defect in red and purple cut flowers, especially in cut chrysanthemum which have a relative long vase life. Here, the effect of sucrose on change in anthocyanin contents during the vase life of 'Dante Purple' cut chrysanthemum was studied. Results showed that 500 mM sucrose as holding solution could significantly delay the decrease in anthocyanin content and maintain the ornamental value for as long as 38 vase days. Moreover, the sucrose also increased the flower diameter, soluble sugar contents and total antioxidant capacity, while decreasing the malondialdehyde contents. Further studies suggested that the transcript levels of anthocyanin biosynthetic genes and transcription factors, *CmMYB6* and *CmMYB#7*, had continuously decreased during the vase life. The changes in these genes expression patterns was retarded by the sucrose treatment, except for *CmMYB#7* which is a repressor of anthocyanin biosynthesis gene expression. The decline in relative expression of *CmMYB#7* was accelerated by sucrose. These results have supplied clues to study the mechanism whereby sucrose serves as a signal molecule to regulate anthocyanin biosynthesis.

## INTRODUCTION

Production and consumption of cut flowers have increased rapidly during recent years. Among the ornamental cut flowers, cut chrysanthemums rank second globally after rose in terms of socioeconomic importance (*Teixeira da Silva et al., 2013*). Cut flowers have multiple uses, such as improving quality of daily life, providing a ceremonial atmosphere, generally producing a symbol of welcome and happiness, *etc*. As the typical ornamental crops, the post-harvest quality and vase life for cut flowers are the main important factors.

Corresponding authors
Lili Xiang, 81975810@qq.com
Fang Li, lifang68@zju.edu.cn

Chrysanthemum belongs to the long vase life group of cut flowers, lasting for 2 to 4 weeks (*Nguyen & Lim, 2021*). This makes the post-harvest quality more important. Floral color is the most intuitive quality and attracts the most attentions of customers, breeders and researchers (*Mekapogu et al., 2020*). However, floral color fading, absence, or breakdown, coupled with the post-harvested senescence, seriously hampers the development of the cut flower industry. The post-harvested fresh condition of cut flowers is maintained by a preservative solution usage to provide fresh flowers with a long vase life to the final customer. Developing effective and eco-friendly treatments to maintain and even improve the floral color during the vase life is the essential objective.

Effects of several different kinds of preservative solution, including sugar, on the vase life of cut flowers have been studied (reviewed by *Nguyen & Lim, 2021*). The application of exogenous soluble sugars in holding solutions can be used to prolong vase life of cut flowers. It has been shown that sucrose could enhance the induction of anthocyanin contents in petals of cut *Eustoma* flowers treated with silver thiosulfate complex (*Shimizu & Ichimura, 2005*). Recently, *Zhang et al. (2022)* found that a supply of glucose could induce *PsMYB2* mediated anthocyanin accumulation in *Paeonia suffruticosa* cut flower. Anthocyanins are one of the main pigments contributing to a broad variety of colors from pink to purple in plants, especially in flowers. The maintainenance or induction of anthocyanin accumulation is useful for preventing the floral color fading. However, the effects and mechanism of sugars on anthocyanin metabolism in cut chrysanthemums have rarely been studied.

Anthocyanin biosynthesis is catalyzed by a series of enzymes in chrysanthemum, including *CmCHS*, *CmCHI* and *CmF3'H* were are expressed more highly at the early flower development stages, and *CmF3H*, *CmDFR*, *CmANS* and *CmUFGT* which are coordinately expressed throughout all stages of ray floret development (*Huang et al., 2013*). The anthocyanin biosynthetic genes are transcriptionally regulated by MYB and bHLH transcription factors (*Hong et al., 2015*; *Xiang et al., 2015*). Furthermore, the MYBs are involved as anthocyanin biosynthesis regulators both as activators and repressors, *e.g.*, *CmMYB6* and *CmMYB#7*, respectively, in chrysanthemums (*Liu et al., 2015*; *Xiang et al., 2019*). The mechanisms which affect anthocyanin metabolism could be traced to the changes in expression patterns of these biosynthetic genes and TFs (reviewed by *Mekapogu et al., 2020*).

To evaluate the effect of sugars on anthocyanin metabolism in cut chrysanthemums, 500 mM sucrose was applied as holding solution for 'Dante Purple' in this study based on preliminary experimental results (Fig. S1) and the previous reports (*Zhang et al., 2015*), which suggested that sucrose had significantly positive effects on prolonging vase life and improving quality of cut chrysanthemums. Compared to the control group, sucrose treatment increased the anthocyanin contents and total antioxidant capacity of petals, while it decreased the malondialdehyde contents. The anthocyanin biosynthetic genes showed decreased profiling during the vase life of 'Dante Purple', while sucrose treatment retarded these changes. The transcript levels of two MYB members, identified as anthocyanin biosynthetic activator (*CmMYB6*) and repressor (*CmMYB#7*) respectively, responded differently to sucrose. Based on these results, the 500 mM sucrose could be

supplied as effective and eco-friendly preservative to improve the post-harvest quality of cut chrysanthemums.

# MATERIALS AND METHODS

## Plant materials

The spray cut chrysanthemum, *Chrysanthemum morifolium* var. Dante Purple, with bright purple petals was selected for this study. The cut flowers at the full-bloom stage with 60 cm length of stems were harvested in January 2021, from Honghua Horticulture Company in Yunnan province, China. The stems were trimmed to 40 cm and all but the upper three to five leaves were removed and rehydrated in distilled water after transportation to the lab.

## Treatments

A total of 1 day after rehydration, the cut flowers were placed in glass flasks filled with 500 mL of 500 mM sucrose solution or distilled water as control (CK). The day was set as the zero vase day. The solution or distilled water was replaced every 6 days. Cut flowers of the same color and size were selected and divided into different groups randomly, with 10 stems in each group. Each treatment contained three different groups as three biological replications.

The cut flowers were cultured under $11 \pm 1$ °C, $55 \pm 5\%$ relative humidity. The light/dark condition (12 h/12 h) was managed by indoor natural sunlight coupled with fluorescent lamps as an artificial light source. From the zero vase day, the status of cut flowers was observed every day of the treatment. Photographs were taken at 0, 6, 12 and 18 vase days. Then the petals located in the $4^{th}$–$6^{th}$ layer of flowers were collected and the color index measured. Finally, these petals were frozen in liquid nitrogen and stored at −80 °C for further studies.

## Flower diameter measurement

The maximum width of each 'Dante Purple' flower was measured using vernier calipers and recorded as flower diameter. The average value of the total 30 stems included in three biological replications of each treatment group was recorded as the flower diameter on vase day 0, 6, 12 and 18, respectively. The rate of flower diameter increase was calculated as $[(D_t-D_0)/D_0] \times 100$, where the $D_t$ was the diameter of the flower on vase day 6, 12 or 18 ($D_0$ was on zero vase day).

## Flower color evaluation

The colors of the upper epidermis of the $4^{th}$–$6^{th}$ layer petals were evaluated using colorimeter MiniScan EZ (HunterLab, Reston, VA, USA). The lightness ($L^*$) and two chromatic components $a^*$ and $b^*$ of the CIE$L^*a^*b^*$ color coordinate were measured in daylight conditions. Chroma ($C^*$) was calculated as $C^* = (a^{*2} + b^{*2})^{1/2}$. The average value of 10 different petals in one flower was used to evaluate this flower's color.

## Soluble sugar contents determination

The content of soluble sugars was measured according to the previous method with some modifications (*Zhang et al., 2022*). Powders of frozen petal samples (200 mg) were

extracted in 6 mL 80% ethanol (v/v) at 80 °C for 30 min. The supernatant was collected after centrifuging at 10,000 g for 5 min and was adjusted into 10 mL. The contents of soluble sugars were determined with the sulfuric acid anthrone method at a wavelength of 620 nm.

## Total antioxidant capacity and malondialdehyde content measurement
Total antioxidant capacity (T-AOC) and malondialdehyde (MDA) content in the petals were measured using total antioxidant capacity assay kit (ABTS method) and Malondialdehyde assay kit (TAB method; Nanjing Jiancheng, Nanjing, China) according to the manufacturer's instructions.

## Anthocyanin contents measurement
Total anthocyanin contents in the petals were measured by the pH difference method as described in our previous report (*Liu et al., 2015*). Anthocyanins were extracted from 1 g frozen powdered samples with methanol/0.05% HCl under 4 °C and dark condition. The absorbances of extracts were measured in a UV-2550 spectrophotometer (Shimadzu, Japan) at 510 and 700 nm. The total anthocyanin content was calculated as $A = (A_{510,1} - A_{700,1}) - (A_{510,5} - A_{700,5})$, where $A_{510,1}$ was the absorbance value of 510 nm at pH 1.0, the rest may be deduced by analogy.

## QPCR (Real-time quantitative PCR) analysis
The total RNAs of the petals in each sample were extracted using RNAprep Pure Plant Kit (Tiangen, China), according to the manufacturer's instructions. The first strand of cDNA was synthesized from 1 μg of total RNA using HiScript® II Q RT SuperMix for qPCR (+gDNA wiper; Vazyme, Xuanwu Qu, China) according to the manufacturer's instructions. QPCR were carried out with AceQ® qPCR SYBR Green Master Mix (Vazyme, China) and the protocol set as 95 °C 5 min, 45 cycles (95 °C 10 s, 60 °C 30 s), for melting curve (95 °C 15 s, 60 °C 60 s, 95 °C 15 s).

The relative expression levels of *CmCHS*, *CmCHI*, *CmF3H*, *CmF3'H*, *CmDFR*, *CmANS*, *CmUFGT*, *CmMYB6* and *CmMYB#7* were normalized using the control gene (*CmActin*). And the relative expression level was analyzed by the $2^{-\Delta Ct}$ method (*Fu et al., 2023*). Primers used in QPCR for the genes were designed with Primer-BLAST (https://blast.ncbi.nlm.nih.gov/Blast.cgi) and are listed in Table 1. No-template reactions were set as negative controls for each gene.

## Statistical analysis
The statistical analysis was performed by one-way analysis of variance (ANOVA), using SPSS software version 24.0. Duncan's multiple range test was employed and differences of $P < 0.05$ were considered significant.

# RESULTS

## Sucrose delayed the flower color fading during the vase life
Compared to the control groups, as well as glucose and mannose treatment groups, flower color of 'Dante Purple' was maintained well by treatment with 200, 500 and 800 mM

**Table 1 Primer sequences for QPCR analysis.**

| Gene | Forward primers(5′–3′) | Reverse primer(5′–3′) |
|---|---|---|
| *CmActin7* | CACCCCCAGAGAGAAAATAC | ATCTGTTGGAAGGTGCTGAG |
| *CmMYB6* | ATGGGGGAGTACAGAAAAATG | TCATAGTTGGTCCGAATTTA |
| *CmMYB#7* | TACAGGATGCACAAGCTTGTTG | ACATCGTATGAGACAAAGTGTC |
| *CmbHLH2* | GTGAAGGTGAAGGGTATTAGGGGG | CTCTTCAAACGTCCTTCACATACC |
| *CmCHS* | CAAGGAGGAGAAGATGAGAG | CCGAACCCGAATAAAACAC |
| *CmCHI* | GCAGGTGTGAGAGGTATG | GCAACGGAATCGCTTTATC |
| *CmF3H* | CACGGGTAATGTTAGGTAGG | TGTAGGTCAAATCGGTCAAG |
| *CmF3'H* | AGGCGGATTCATCGTTTC | ACTCTTTGGGCTTATCAGG |
| *CmDFR* | TAGTAACAAAGGCGGACAC | GGATTATTCACCAAGTATGCTC |
| *CmANS* | GGGCTCCAACTACTCTATG | TCCTAACCTTCTCCTTATTCAC |
| *CmUFGT* | ATCACAGGGACTATCAACC | TCCACCACCAGACAACTA |

sucrose (Fig. S1). Furthermore, the 500 mM sucrose treatment had the most excellent effects on improving the postharvest qualities through promoting the flower growth during the vase life (Fig. S1). Thus, the 500 mM sucrose treatment was chosen in the subsequent studies.

At the zero vase day after one day rehydration, flowers showed no difference between the control and 500 mM sucrose treated groups (Fig. 1A). Both lingual and tubular flowers were found in the spray cut chrysanthemum and showed bright and dark purple color, respectively. The tubular flowers had developed into lingual ones gradually during the vase lives. The flower diameters continuously increased in both control and sucrose treated groups but to significantly different extents (Fig. 1B). In the control groups, the flower diameters increased more than 15% at six vase days compared to the zero vase day, and they reached nearly 25% at 18 vase days (Fig. 1B). When the flowers were cultured with sucrose, the diameters extended to more than 25% and 35% at six and 18 vase days respectively (Fig. 1B).

Color fading was observed in the flowers cultured with distilled water, and the purple color become lighter and lighter during the 18 days vase lives (Fig. 1A). Correspondingly, the L* index, which indicated the lightness of the petals, increased dramatically from about 35 at 0 vase day to more than 50 at 18 vase days in control groups (Fig. 2A). These significant increases in L* were effectively retarded and reached only 42 at 18 vase days in the sucrose treated group.

Compared to the control, the flower colors were effectively improved by sucrose treatment (Fig. 1A). All of the reduction in redness (a*), yellowness (b*) and chroma (C*) in petals had been significantly delayed with sucrose solution culture during the vase lives of cut flowers (Figs. 2B–2D).

## Sucrose improved the post-harvest quality of cut chrysanthemum

Besides retarding the color fading, another significant effect of sucrose on cut chrysanthemum was the improvement of the post-harvest quality. Compared to the

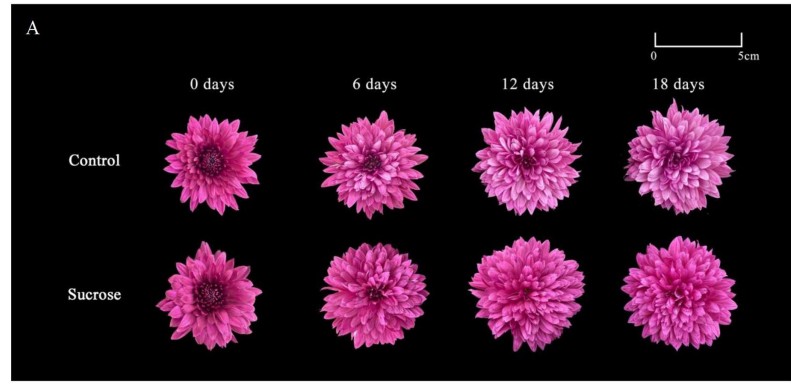

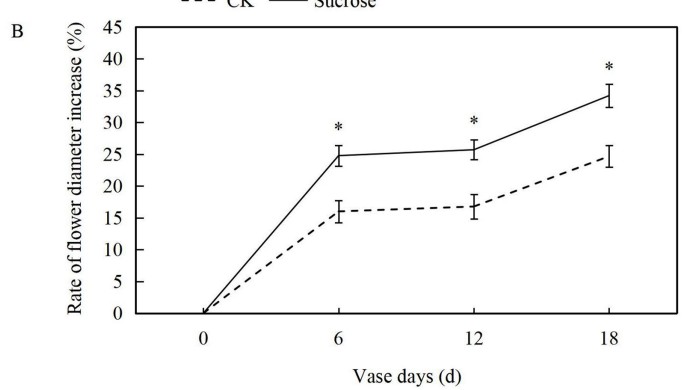

**Figure 1 Effect of Sucrose on flower colour (A) and diameter (B) during the vase life of cut Chrysanthemum 'Dante Purple'.** Error bars indicate ±SEM from three biological replicates and black asterisks represent significant differences between control and treatment groups (*$P < 0.05$).

control, flowers cultured with sucrose were much more ornamental even at 38 vase days (Fig. S2).

To evaluated the physiological characteristics of the cut flowers, several indexes including contents of soluble sugar, malondialdehyde, and anthocyanin, as well as total antioxidant capacity in petals were measured. These physiological indices showed different changing patterns during the vase lives (Fig. 3). The soluble sugar contents in the petals were nearly 20 mg/g and remained stable in the control group, while they were dramatically increased by sucrose and reached to about 60 mg/g at the 18 vase days (Fig. 3A). The total antioxidant capacities in chrysanthemum showed slight increases in the control group. And they were significantly enhanced by sucrose treatment, to around 28–53% higher during the vase lives, even with a weak decline at 18 vase days compared to the six and 12 vase days (Fig. 3B).

Contents of malondialdehyde in the petals showed a decrease from zero to 12 vase days and increase at 18 vase days in both control and treated groups (Fig. 3C). However, the degree of decline in control groups were much slighter than the treated flowers. Compared to the 0 vase day level, contents of malondialdehyde decreased from 0.20 to 0.175 nmol/mL (six vase days), and increased to 0.225 nmol/mL (18 vase days) in control groups. While in

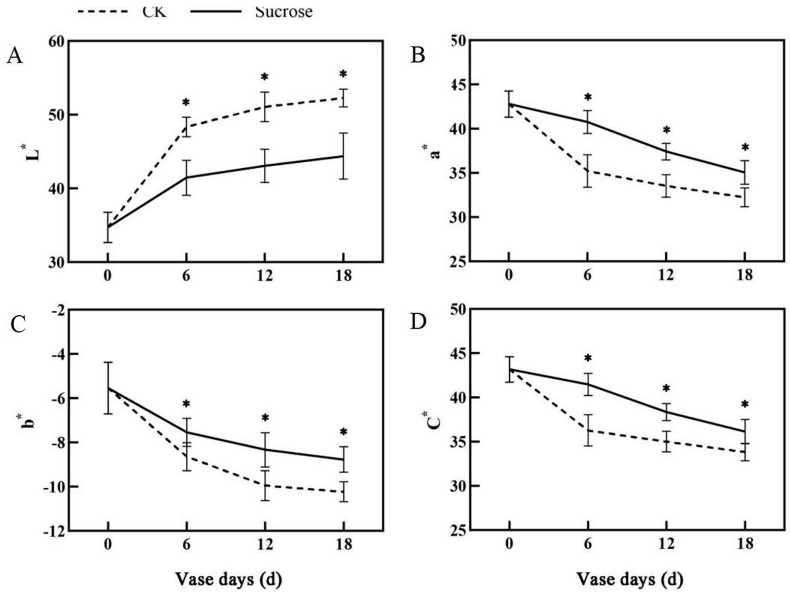

**Figure 2 Changes of the CIEL\*a\*b\* color coordinates between the sucrose treatment group and the control (CK) during the vase life of cut flowers.** The L\* indicated lightness, a\* redness, b\* yellowness and C\* meant chroma, respectively. Error bars indicate ±SEM from three biological replicates and black asterisks represent significant differences between control and treatment groups (*P < 0.05).

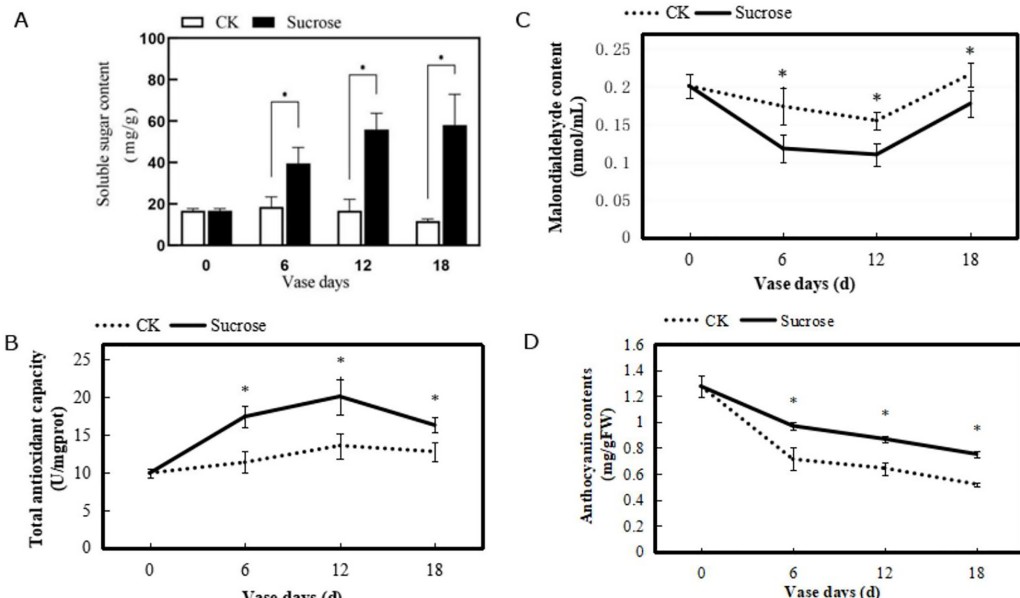

**Figure 3 Changes of the contents of soluble sugar (A), the total antioxidant capacity (B), contents of malondialdehyde (C) and anthocyanin (D) of petals during the vase life of cut flowers.** Error bars indicate ±SEM from three biological replicates and black asterisks represent significant differences between control and treatment groups (*P < 0.05).

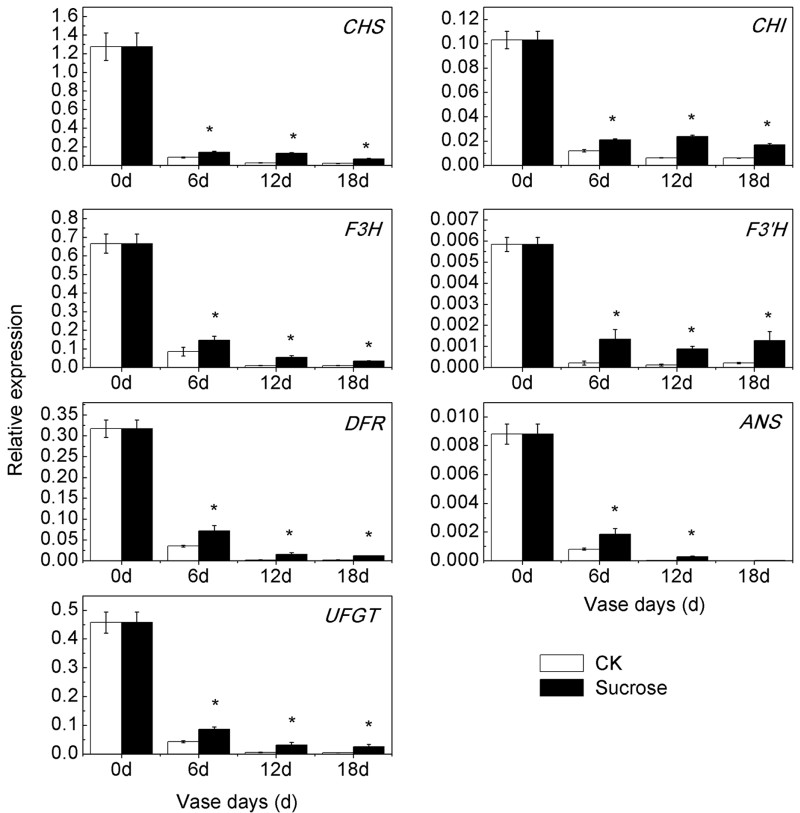

**Figure 4 Effect of sucrose on the relative expression patterns of anthocyanin biosynthetic genes during the vase life.** Error bars indicate ±SEM from three biological replicates and black asterisks represent significant differences between control and treatment groups ($^*P < 0.05$).

the sucrose treated groups, their contents decreased from 0.20 to 0.125 nmol/mL (six vase days), and 0.175 nmol/mL (18 vase days) (Fig. 3C).

The anthocyanin contents in petals of both control and sucrose treated groups decreased during the vase lives (Fig. 3D). However, a sharper reduction was detected in the control groups, where the anthocyanin contents decreased from 1.3 mg/gFW (zero vase day) to 0.7 mg/gFW (six vase days) and 0.5 mg/gFW (18 vase days). This reduction has been delayed with sucrose treatment and the anthocyanin contents were still maintained at 0.8 mg/gFW after 18 vase days (Fig. 3D).

## Sucrose retarded the decrease of anthocyanin biosynthetic genes expression patterns

Consistent with the reduction of anthocyanin content during the vase lives, the transcript profiling of seven anthocyanin biosynthetic genes showed significant decreases in both control and sucrose treated groups (Fig. 4). However, the down-regulation patterns occurred to different extents in control and sucrose treated groups. Compared to zero vase day, the relative transcript levels of *CmCHS* decreased to 6.4% (six vase days), to 2% (12 and 18 vase days) in control groups, while they were reduced to 12% (six vase days), to 10% (12 vase days), to 6% (18 vase days) in sucrose treated groups (Fig. 4). Similar changes

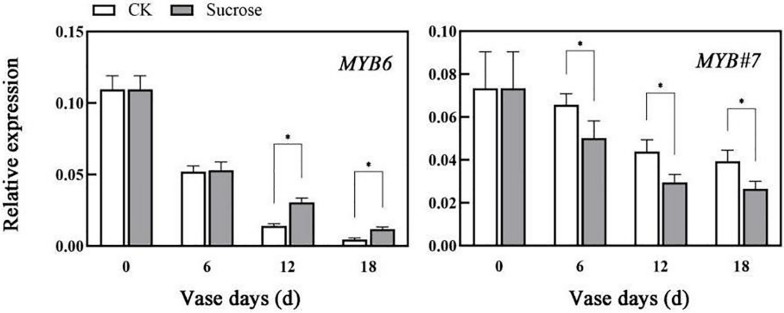

**Figure 5 Effect of sucrose on the relative expression patterns of two MYBs, which were verified as anthocyanin biosynthesis activator (*MYB6*) and repressor (*MYB#7*) respectively.** Error bars indicate ±SEM from three biological replicates and black asterisks represent significant differences between control and treatment groups (*$P < 0.05$).

existed in expression patterns of *CmCHI*, *CmF3H*, *CmDFR*, *CmANS* and *CmUFGT* (Fig. 4). The transcript levels of *CmF3'H* sharply decreased to 0.4% during the vase lives in the control group, while it was maintained at 2% in the sucrose treated groups (Fig. 4). Taken together, sucrose could significantly retard the decrease of anthocyanin biosynthetic genes expression patterns during the vase lives of cut chrysanthemum.

## Sucrose had opposite effects on activator-/repressor- MYB members

The relative expression patterns of both *CmMYB6* and *CmMYB#7* decreased during the vase lives of cut flowers (Fig. 5). Sucrose could delay the decrease in transcripts of *CmMYB6*, which is an activator of anthocyanin biosynthesis regulation (*Liu et al., 2015*). Compared to the control groups, the relative transcript levels of *CmMYB6* decreased to 50% (six vase days), 15% (12 vase days), to 2% (18 vase days) in control groups, while they reduced to 50% (six vase days), 25% (12 vase days), to 10% (18 vase days) in sucrose treated groups (Fig. 5). However, the decreasing patterns of *CmMYB#7* transcripts, which acts as repressor in anthocyanin accumulation were significantly enhanced by sucrose treatment (Fig. 5).

## DISCUSSION

To prolong vase life and improve the post-harvested quality of cut flowers, preservative solutions containing sugars, germicides or biocides, salts and growth regulators, have generally become the best choice in the floral industry (*Nguyen, Jung & Lim, 2020*). Effects of different constituents on the post-harvested cut flowers had been studied. In cut *Eustoma* flowers, the positive effects on quality and vase life of pulse treatment with silver thiosulfate complex (STS), 4% sucrose and their combination had been verified (*Shimizu & Ichimura, 2005*). *Rezvanypour & Osfoor (2011)* found that 0.5 mM silver thiosulfate and sucrose combination enhanced the water uptake by cut roses. The different effects on delaying senescence in cut lily of PGRs, including salicylic acid, citric acid, gibberellic acid and clove oil, had been analyzed after pulsing with pre-optimized sucrose 5% (*Aziz et al., 2020*). Taken together, sugar is usually used as the basal constituent in the preservative solution. As the gold key of floral energy sources, sugar (usually sucrose) has positive

effects on bud opening, flower size increase, flower color production, vase life extension, *etc*.

As a signaling molecule, sucrose controls many physiological activities and metabolic reactions in plants, such as carbohydrate metabolism, accumulation of storage proteins, sucrose transport, floral induction, and anthocyanin accumulation (*Yoon et al., 2021*). Several studies had found that sucrose played positive effects on anthocyanin biosynthesis. *Arabidopsis* germinated or grown on a sugar-containing medium accumulated high levels of anthocyanins (*Qiu et al., 2014*; *Rolland, Baena-Gonzalez & Sheen, 2006*). *Hase et al. (2010)* verified that petunia seedlings treated with 3% sucrose could accumulate significant amount of anthocyanin. A similar phenomenon has been reported in torenia (*Torenia fournieri*) leaves (*Naing et al., 2021*) and postharvest peaches (*Prunus persica*) peel (*Tian et al., 2022*). Compared to glucose and fructose, sucrose is the most effective and specific activator of anthocyanin biosynthesis in *Arabidopsis* seedlings (*Solfanelli et al., 2006*). This sucrose-induced anthocyanin accumulation can be helpful to improve post-harvested quality of cut flowers, especially for cut chrysanthemum whose flower color fading is the prominent defect. Although no sucrose-induced anthocyanin accumulation had been detected in this study, the significant delaying of flower color fading by sucrose supplied the economical and eco-friendly preservation treatment for the cut flowers industry.

At present, the detailed information of sugars serving as signal molecules to regulate anthocyanin biosynthesis largely has not been revealed (*Zhang et al., 2022*). With the most direct relationship to anthocyanin accumulation, the biosynthetic genes and transcript factors, including *MYBs*, had been up-regulated during the sugar-induced anthocyanin biosynthesis in *Arabidopsis*, petunia, grape and radish (*Hara et al., 2003*; *Neta-Sharir, Shoseyov & Weiss, 2000*; *Solfanelli et al., 2006*; *Zheng et al., 2009*). The expression patterns of these biosynthetic and transcription regulator genes also responded to sucrose treatment here, and the changes of their transcript levels were consistent with the anthocyanin contents in cut flowers (Figs. 3D, 4 and 5). Among the different transcript factor members, MYBs showed the most specific regulating effects on anthocyanin accumulation (*Wang et al., 2022*). MYBs could be divided into activators and repressors according to their regulatory effects on transcription. In our previous studies, *CmMYB6* and *CmMYB#7* have been verified as activator and repressor respectively in chrysanthemum (*Liu et al., 2015*; *Xiang et al., 2019*). Compared to the repressor, the activator more strongly affected during sucrose-induced anthocyanin accumulation and the changes in *CmMYB6* transcript levels were much sharper and significant (Fig. 5).

To uncover the mechanism of how sugars serve as signal molecules to regulate anthocyanin biosynthesis, it would be helpful to study the relationship between these anthocyanin related genes and sucrose-responsive genes, such as *sucrose synthase* (*SUS4*) and *sucrose transporter* (*SUT4*).

## ACKNOWLEDGEMENTS

We sincerely thank Prof. Donald Grierson from Zhejiang University and University of Nottingham for his insightful comments on the manuscript.

### Funding

This research was supported by the National Key Research and Development Program (2018YFD1000405), the National High Technology Research and Development Program of China (2013AA102700), and the National Natural Science Foundation of China (32202533). The funders had no role in study design, data collection and analysis, decision to publish, or preparation of the manuscript.

### Grant Disclosures

The following grant information was disclosed by the authors:
National Key Research and Development Program: 2018YFD100040.
National High Technology Research and Development Program of China: 2013AA102700.
National Natural Science Foundation of China: 32202533.

### Competing Interests

The authors declare that they have no competing interests.

### Author Contributions

- Xiao-fen Liu performed the experiments, analyzed the data, prepared figures and/or tables, and approved the final draft.
- Ruping Teng performed the experiments, analyzed the data, prepared figures and/or tables, and approved the final draft.
- Lili Xiang conceived and designed the experiments, analyzed the data, prepared figures and/or tables, authored or reviewed drafts of the article, and approved the final draft.
- Fang Li conceived and designed the experiments, prepared figures and/or tables, authored or reviewed drafts of the article, and approved the final draft.
- Kunsong Chen analyzed the data, authored or reviewed drafts of the article, and approved the final draft.

### Data Availability

  The raw measurements are available in the Supplemental Files

### Supplemental Information

Supplemental information for this article can be found online at http://dx.doi.org/10.7717/peerj.16520#supplemental-information.

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
