# Peer review of "Sucrose-delaying flower color fading associated with delaying anthocyanin accumulation decrease in cut chrysanthemum"

_PeerJ, doi:10.7717/peerj.16520_

## Round 0.1 · original submission · Minor Revisions

This is an interesting study with findings that could have a positive impact on the cut flower industry. Please revise the manuscript based on the reviewers' recommendations, addressing their comments point by point.

Reviewer 1 ·

Basic reporting

In this article, the authors reported 500 mM sucrose treatment could significantly delay the petal color fading of chrysanthemum. Also, they found that the treatment of sucrose increased the flower diameter, soluble sugar contents and total antioxidant capacity, while decrease the malondialdehyde contents. Furthermore, they found that the decrease in transcripts of anthocyanin-associated structural genes and TFs were retarded by the sucrose treatment, except for a repressor in anthocyanin biosynthesis regulation: CmMYB#7. The results demonstrated by the authors are of interesting and have application values in production. Below I raise some points.
1. Why did the authors choose 500 mM as the concentration of sucrose treatment?
2. The authors should check the transcriptions of genes in the sucrose signaling pathway to serve as positive controls of sugar treatment.
3. Also, the transcriptions of senescence-associated genes should be checked.

Experimental design

no comments

Validity of the findings

no comments

Additional comments

no comments

·

Basic reporting

In title Sucrose-delaying Flower color fading associated with delaying anthocyanin accumulation decrease in cut chrysanthemum it includes flower color fading is the prominent defect in red and purple cut flowers, especially in cut chrysanthemum which have a relative longer vase life. According to my opinion recommended minor revision. Some aspects of the work need to be revised.
1. Poor written English
2. Avoid using keywords already present in the title of the study.
3. Line 30-31 rewrite or improve this sentence.
4. In section Flower diameter detection methodology should be mentioned in detail.
5. In Introduction section with the potential field implications of the research work.
6. Gene name should be italic, including title and whole manuscript.
7. The methodology of qRT-PCR should be in detail.
8. Did you assess some practical implications of the genes and their expression?
9. More references to support the finding.
10. The figures and tables were informative. They just need better captions.
11. In qRT-PCR validation section should be mentioned results in detail.

Experimental design

no comments

Validity of the findings

nill

Additional comments

nill

---

## Round 0.2 · Minor Revisions

Thanks for submitting the revised manuscript and rebuttal letter. I have carefully reviewed your submission and I have some feedback for you.

English language: I agree with reviewer 2 that the English language of the manuscript is not yet of a publishable standard. There are many small grammar problems, many of which were marked by Microsoft Word language tools. I would recommend that you get help from a professional language editing service to improve the English language of your manuscript.

Point 5 from reviewer 2: I found that you did not adequately address this comment in your revised manuscript. In the rebuttal letter, you state that this point has been "corrected," but I do not see any changes to the manuscript that address this point. I would recommend that you revise your manuscript to specifically address reviewer 2's concerns about point 5.

Points 7-9 from reviewer 2: Similar problems. I would recommend that you carefully review these points and make any necessary revisions to your manuscript.

I would like to see a revised manuscript that addresses all of the concerns raised by the reviewers. Once you have made the necessary revisions, please resubmit your manuscript for further consideration.

**Language Note:** The Academic Editor has identified that the English language must be improved. PeerJ can provide language editing services - please contact us at [email protected] for pricing (be sure to provide your manuscript number and title). Alternatively, you should make your own arrangements to improve the language quality and provide details in your response letter. – PeerJ Staff

---

## Round 0.3 · accepted · Accept

Thank you for your careful revision of the manuscript. I am glad to find that the English language has been improved significantly and now meets the journal's standards for publication.

Congratulations! I look forward to seeing your work in print.

The Section Editor recommends that you add a "Conclusions" section.

Reviewer 1 ·

Basic reporting

no comment

Experimental design

no comment

Validity of the findings

no comment

Additional comments

The manuscript can be accepted now.

·

Basic reporting

The article meets the PeerJ criteria and should be accepted as is

Experimental design

N/A

Validity of the findings

N/A

Additional comments

N/A